TVA-based modeling of short-term memory capacity, speed of processing and perceptual threshold in chronic stroke patients undergoing cognitive training: case-control differences, reliability, and associations with cognitive performance

Richard Geneviève genevieve.richard@medisin.uio.no 1 2 3
Petersen Anders 4
Ulrichsen Kristine Moe 1 2 3
Kolskår Knut K. 1 2 3
Alnæs Dag 1
Sanders Anne-Marthe 1 2 3
Dørum Erlend S. 1 2 3
Ihle-Hansen Hege 5
Nordvik Jan E. 6
Westlye Lars T. l.t.westlye@psykologi.uio.no 1 2 7
1 NORMENT, Division of Mental Health and Addiction, Oslo University Hospital & Institute of Clinical Medicine, University of Oslo , Oslo , Norway
2 Department of Psychology, University of Oslo , Oslo , Norway
3 Sunnaas Rehabilitation Hospital HT , Nesodden , Norway
4 Center for Visual Cognition, Department of Psychology, University of Copenhagen , Copenhagen , Denmark
5 Department of Geriatric Medicine, Oslo University Hospital , Oslo , Norway
6 CatoSenteret Rehabilitation Center , Son , Norway
7 KG Jebsen Centre for Neurodevelopmental Disorders, University of Oslo , Oslo , Norway
Rocha Joao
Electronic publication date: 2020 Oct 28
Publication date: 2020
Volume: 8
Electronic Location ID: e9948
Received 2019 Aug 27; Accepted 2020 Aug 25
Copyright: ©2020 Richard et al.
Copyright year: 2020
Copyright holder: Richard et al.
License: This is an open access article distributed under the terms of the Creative Commons Attribution License, which permits unrestricted use, distribution, reproduction and adaptation in any medium and for any purpose provided that it is properly attributed. For attribution, the original author(s), title, publication source (PeerJ) and either DOI or URL of the article must be cited.
License URL: https://creativecommons.org/licenses/by/4.0/

Keywords: Theory of visual attention, Computerized cognitive training, Attentional deficits, Transcranial direct current stimulation, Cerebral stroke, Longitudinal assessment

Funding: The Norwegian ExtraFoundation for Health and Rehabilitation 2015/FO5146 The Research Council of Norway 249795 248238 298646 300768 The South-Eastern Norway Regional Health Authority 2014097 2015044 2015073 2019101 The European Research Council under the European Union’s Horizon 2020 research and innovation program ERC Starting Grant 802998 Sunnaas Rehabilitation Hospital The Department of Psychology, University of Oslo This study was supported by the Norwegian ExtraFoundation for Health and Rehabilitation (2015/FO5146), the Research Council of Norway (249795, 248238, 298646, 300768), the South-Eastern Norway Regional Health Authority (2014097, 2015044, 2015073, 2019101), the European Research Council under the European Union’s Horizon 2020 research and innovation program (ERC Starting Grant 802998), Sunnaas Rehabilitation Hospital, and the Department of Psychology, University of Oslo. The funders had no role in study design, data collection and analysis, decision to publish, or preparation of the manuscript.

==============================
Attentional deficits following stroke are common and pervasive, and are important predictors for functional recovery. Attentional functions comprise a set of specific cognitive processes allowing to attend, filter and select among a continuous stream of stimuli. These mechanisms are fundamental for more complex cognitive functions such as learning, planning and cognitive control, all crucial for daily functioning. The distributed functional neuroanatomy of these processes is a likely explanation for the high prevalence of attentional impairments following stroke, and underscores the importance of a clinical implementation of computational approaches allowing for sensitive and specific modeling of attentional sub-processes. The Theory of Visual Attention (TVA) offers a theoretical, computational, neuronal and practical framework to assess the efficiency of visual selection performance and parallel processing of multiple objects. Here, in order to assess the sensitivity and reliability of TVA parameters reflecting short-term memory capacity (K), processing speed (C) and perceptual threshold (t0), we used a whole-report paradigm in a cross-sectional case-control comparison and across six repeated assessments over the course of a three-week computerized cognitive training (CCT) intervention in chronic stroke patients (> 6 months since hospital admission, NIHSS ≤ 7 at hospital discharge). Cross-sectional group comparisons documented lower short-term memory capacity, lower processing speed and higher perceptual threshold in patients (n = 70) compared to age-matched healthy controls (n = 140). Further, longitudinal analyses in stroke patients during the course of CCT (n = 54) revealed high reliability of the TVA parameters, and higher processing speed at baseline was associated with larger cognitive improvement after the intervention. The results support the feasibility, reliability and sensitivity of TVA-based assessment of attentional functions in chronic stroke patients.

Introduction

Attentional deficits following stroke are common, pervasive and persistent (Barker-Collo et al., 2010a), likely due to the distributed functional neuroanatomy supporting the range of attentional sub-functions (Rosenberg et al., 2017). Specific functions of attention, such as the ability to rapidly detect changes in perceptual scenes, or internally sustain focus over several stimuli over an extended period, are fundamental to more complex operations supporting everyday functions such as learning, social interactions and cognitive performance in general, and are important predictors for functional recovery in stroke patients (Peers et al., 2020). For instance, attentional functions assessed at hospital discharge has been shown to be relevant for predicting future recovery (Hyndman, Pickering & Ashburn, 2008), sustained visual and auditory attention measured two months after stroke was a strong predictor of long-term motor recovery (Robertson et al., 1997), poorer attentional performance was associated with a more negative impact of stroke on daily functioning (McDowd et al., 2003), and attentional abilities have been associated with language recovery after stroke (Geranmayeh, Brownsett & Wise, 2014).

Whereas the prevalence of attentional deficits following stroke is high, the reported estimates vary depending on assessment tool (Barker-Collo et al., 2010b; Xu et al., 2013), and subtle deficits are most likely underestimated when based on traditional bedside examination (Rinne et al., 2013). Given the high prevalence of attentional impairments in the acute and chronic stages of stroke and the relevance of attentional functions as a predictor of recovery and everyday functions, there is a need to identify specific and reliable behavioral markers of attentional abilities in individual patients. From a cognitive perspective, visual attention broadly refers to the joint set of cognitive processes that enables efficient and continuous selection and discrimination between competing stimuli from visual scenes of various degrees of complexity, including attentional capacity which comprises speed and storage aspects, and is a core determinant of cognitive performance in general. The cognitive and computational sub-processes involved can be operationalized and assessed using various paradigms. Representing one of the most comprehensive and coherent accounts of attentional capacity, the Theory of Visual Attention (TVA; Bundesen, 1990) proposes a number of computational parameters that have been shown to be sensitive to cognitive aging (Espeseth et al., 2014; Habekost et al., 2013; Wiegand et al., 2018), as well as attentional impairments in several brain disorders, including stroke (Habekost, 2015; Habekost & Starrfelt, 2009). Briefly, TVA offers a theoretical, computational and practical framework to assess an individual’s efficiency of visual selection performance and parallel visual processing of multiple objects. In TVA-based assessments, participants have to either report as many letters as possible (whole-report) or only a subset (partial-report) from a set of briefly displayed letters. TVA assumes that the correctly reported letters are the winners of a race among all the letters in the visual field (later referred to as a biased competition model; Desimone & Duncan, 1995). Moreover, it assumes that the course of the visual encoding process depends on five distinct mathematical parameters: perception threshold (t0), visual processing speed (C), visual short-term memory capacity (K); visual distractibility (α) and relative attentional weight of each visual object (w) (Habekost, 2015). While correlations have been reported between short-term memory capacity (K) and visual processing speed (C) (Habekost, Petersen & Vangkilde, 2014), previous studies have demonstrated empirical independence and generally high reliability of the five parameters derived from TVA, with short-term memory capacity (K) consistently showing highest reliability (Finke et al., 2005; Habekost & Bundesen, 2003; Habekost, Petersen & Vangkilde, 2014; Habekost & Rostrup, 2006).

In addition to its attractive psychometric properties, TVA-based parameters are sensitive to attentional dysfunction in clinical groups. Processing speed and short-term memory capacity as measured with TVA have been shown to be selectively impaired in patients with parietal lesions and preserved in patients with frontal lesions (Bublak et al., 2005; Peers et al., 2005). Further, a TVA-based paradigm was shown to be sensitive to asymmetric visual perception after right sided lesions in a group of patients showing minor to no clinical deficits (Habekost & Rostrup, 2006). Moreover, while right hemisphere lesions did not cause deficits in short-term memory capacity, intact white matter connectivity was important for preserved visual short-term memory capacity and ipsilesional processing speed (Habekost & Rostrup, 2007). Together these previous studies highlight the sensitivity of TVA parameters to detect dissociable and subclinical attentional deficits.

Here, we combined a cross-sectional case-control comparison in 70 chronic stroke patients who suffered mild to moderate stroke (>6 months since hospital admission, National Institute of Health Stroke Scale (NIHSS; Lyden et al., 2009) ≤ 7 at hospital discharge) and healthy controls (n = 140) with a longitudinal assessment during the course of a computerized cognitive training (CCT), in which 54 of the stroke patients completed an intensive CCT program and were either assigned to an active transcranial direct current stimulation (tDCS) or sham-tDCS condition in order to test for putative beneficial effects of brain stimulation in combination with CCT (Kolskår et al., 2020). We assessed the sensitivity and reliability of TVA parameters reflecting short-term memory capacity (K), processing speed (C) and perceptual threshold (t0) based on a whole-report behavioral paradigm. In addition, we tested for associations between TVA parameters of visual attention and various clinical measures, including severity, stroke subtype, and lesion extent and location.

Based on the notion that TVA provides sensitive and specific measures of visual attention and that subtle attentional deficits are common and pervasive following even relatively mild strokes, we hypothesized that (1) chronic stroke patients at the group level would show evidence of lower short-term memory capacity, lower processing speed, and higher perceptual threshold compared to healthy peers. Further, assuming that the TVA paradigm provides sensitive measures of subtle attentional deficits (Bublak et al., 2005; Habekost & Rostrup, 2006; Habekost & Rostrup, 2007), we anticipated (2) associations between TVA parameters of visual attention and various clinical measures. More specifically, we hypothesized that higher NIHSS scores would be associated with poorer TVA performance, and tested for associations between TVA parameters and TOAST classification and lesion location. Given that the working memory parameter K consistently has been demonstrated to show high reliability and sensitivity to clinical conditions (Finke et al., 2005; Habekost & Bundesen, 2003; Habekost, Petersen & Vangkilde, 2014; Habekost & Rostrup, 2006), we anticipated stronger association with clinical measures for that parameter. Next, we hypothesized that (3) TVA parameters would constitute reliable and sensitive measures of specific attentional functions in chronic stroke patients in a longitudinal context, with the highest reliability found for parameter K. Further, based on the concept of cognitive reserve (Shin et al., 2020) and assuming that specific attentional abilities are beneficial for learning and cognitive plasticity, we hypothesized that (4) higher attentional abilities as measured using TVA at baseline would be associated with larger cognitive improvement during the course of the CCT, both for patients receiving active and sham tDCS as part of the intervention protocol. Lastly, we hypothesized that (5) improvement in TVA performance would be associated with improvement in CCT performance. Based on previous reports showing strongest practice effect for parameter C (Habekost, Petersen & Vangkilde, 2014), we expected strongest correlation between improvement in CCT performance score and processing speed.

To test these hypotheses, we obtained data using a whole-report behavioral paradigm from chronic stroke patients who were invited to take part of a randomized, double blind study aimed to test the utility of tDCS in combination with CCT to improve cognitive performance following stroke (Kolskår et al., 2020; Richard et al., 2020; Ulrichsen et al., 2020). We used cross-sectional data from 70 chronic stroke patients collected during the first assessment and 140 matched controls, as well as longitudinal data (6 sessions) from 54 chronic stroke patients who completed the full study protocol. As tDCS manipulation (sham vs active group) played a central part in the study design, tDCS condition were taken into account in all models.

Material and Methods

Table 1 summarizes key demographics and clinical information for the healthy controls and the patient group, including the Mini-Mental Status Examination (MMSE; Strobel & Engedal, 2008), the Montreal Cognitive Assessment (MoCA; Nasreddine et al., 2005), the General Anxiety Disorder-7 (GAD-7; Spitzer et al., 2006) and the Patient Health Questionnaire-9 (PHQ-9; Kroenke, Spitzer & Williams, 2001). Figure 1 depicts a schematic timeline of the study protocol.

Table 1 Demographics, descriptive statistics and patients sample characteristics.

	Healthy Controls	Stroke patients
Baseline	Case-Control comparisons	Stroke patients
longitudinal (1st session)	
	Mean (SD)	Range	Mean (SD)	Range	t-test (t(p))	Mean (SD)	Range	
Total N (% females)	140 (39.3%)	–	70 (28.6%)	–	2.33 (.127)c	54 (25.9%)		
Age	67.4 (9.1)	31–81	67.7 (10.1)	24.3–81.8	−0.17 (.865)	69.72 (7.46)	47.8–82.0	
Education	15.92 (3.22)	6–23.5	14.27 (3.78)	7–30	3.04 (.003)	14.38 (3.75)	9–30	
MMSE	28.74 (1.41)	23–30	27.84 (2.04)	22–30	3.23 (.002)	28.00 (1.87)	22–30	
MoCA	27.04 (2.12)	17–30	25.59 (3.16)	14–30	3.42 (.001)	25.92 (2.77)	17–30	
GAD-7	2.09 (2.85)	0–20	2.61 (2.96)	0–12	−1.23 (.220)	2.49 (2.74)	0–11	
PHQ-9	3.21 (3.09)	0–15	4.69 (4.36)	0–17	−2.53 (.013)	4.36 (4.18)	0–17	
TVA parameters					lm (t(p))			
K (letters)	3.07 (0.73)	1.48–5.53	2.72 (0.74)	1.41–4.96	−3.21 (.002)*			
C (Hz)	27.84 (14.84)	6.00–99.33	21.42 (11.50)	3.94–83.39	−3.32 (.001)*			
t0(ms)	25.20 (15.46)	0–79.75	32.41 (20.28)	0–112	3.22 (.002)*			
Error rate	0.11 (0.09)	0–0.46	0.09 (0.08)	0–0.34	−1.68 (.095)			
Patient Characteristics			Cross-sectional (N = 70)			Longitudinal (N = 54)		
Months since stroke	26.67 (9.13)	6–45		25.74 (9.17)	6–45	
NIHSSa	1.31 (1.52)	0–7		1.33 (1.53)	0–7	
TOAST classification for ischemic strokeb	Large artery artherosclerosis (23)
Cardioembolism (7)
Small vessel occlusion (21)
Other (17)		Large artery artherosclerosis (20)
Cardioembolism (6)
Small vessel occlusion (18)
Other (10)	
Stroke location	Right (30)
Left (22)
Brain stem / Cerebellum (9)
Bilateral (7)		Right (23)
Left (18)
Brain stem / Cerebellum (8)
Bilateral (5)	
Notes.

a NIHSS score at hospital discharge.

b One patient had intracerebral hemorrhage. (Kolskår et al., 2020; Richard et al., 2020; Ulrichsen et al., 2020).

c Chi square statistics.

* Significant after Bonferroni correction.

SD = standard deviation; K = short-term memory capacity; C = perceptual processing speed; t0 = perceptual threshold; MMSE = Mini-Mental Status Examination; MoCA = Montreal Cognitive Assessment; GAD-7 = General Anxiety Disorder-7; PHQ-9 = Patient Health Questionnaire-9; lm = linear model.

Figure 1 Schematic timeline of the study protocol.

Assessment refers to the three main cognitive assessment sessions performed prior to and following the intervention. Waiting period refers to the period between the first and second assessment without any active intervention. TVA: Theory of visual attention. tDCS: transcranial direct current stimulation.

Healthy controls

Healthy individuals were recruited through advertisement in newspapers, social media and word-of-mouth (Dørum et al., 2019; Richard et al., 2018). Exclusion criteria included history of stroke, dementia, or other neurologic and psychiatric diseases, alcohol- and substance abuse, medications significantly affecting the nervous system and contraindications for MRI. From a pool of 301 healthy controls who completed the behavioral paradigm and were within the relevant age range (24–81 years), we selected 140 individuals (age 31–81, mean = 67.4, SD = 9.1, 55 females) matched by age and sex using the function matchit with the default method nearest from the R package MatchIt (Ho et al., 2011). Here, we employed a ratio of 2:1, in which 2 healthy controls were selected for each patient.

Patient sample

Patients admitted to the Stroke Unit at Oslo University Hospital and at Diakonhjemmet Hospital, Oslo, Norway during 2013–2016 were invited to participate in a study with the main aim to test the clinical feasibility of combining CCT and tDCS to improve cognitive function in chronic stroke patients (Kolskår et al., 2020; Richard et al., 2020; Ulrichsen et al., 2020). Stroke was defined as any form of strokes of ischemic or hemorrhagic etiology. We included patients in the chronic stage defined as a minimum of 6 months since hospital admission. Exclusion criteria included transient ischemic attacks (TIA), MRI contraindications and other neurological diseases diagnosed prior to the stroke. Clinical severity was indexed by NIHSS scores at the hospital discharge, stroke subtype was classified using the Trial of Org 10172 in Acute Stroke Treatment (TOAST; Adams et al. 1993) classification system, and a coarse four-class classification of lesion location (left hemisphere, right hemisphere, bilateral and brain stem/cerebellum lesions). NIHSS have been shown to have a good predictive values for the prognosis of patients with acute cerebral infarction (Zhao et al., 2018; however, it has demonstrated limited ability to identify cognitive deficits in acute stroke (Abzhandadze, Reinholdsson & Sunnerhagen, 2020). TOAST has been shown sensitive to lesion characteristics (Kang et al., 2003); however, there is a lack of studies assessing TOAST sensitivity to cognitive functions and attentional deficits. None of the patients included in this study reported severe visual or linguistic deficits.

Seventy-two patients completed TVA-based test at the first assessment and 54 of these patients completed the full protocol; including three MRI brain scan sessions, three main cognitive assessment sessions, which included a broad selection of cognitive tests in addition to the TVA paradigm, one EEG assessment session, and 17 CCT sessions. Beyond the three main cognitive assessment sessions, TVA assessment was repeated weekly during the course of the CCT period. Here, we excluded one patient based on incomplete baseline cognitive assessment (first main assessment) and one due to lack of confirmed stroke resulting in the inclusion of 70 patients for the case-control comparisons (age = 24–81, mean = 67.7, SD = 10.1, 20 females). Of these, all patients who completed the six TVA sessions were included in the longitudinal analysis (N = 54, age = 47–82, mean = 69.7, SD = 7.5, 14 females).

The study was approved by the Regional Committee for Medical and Health Research Ethics South-East Norway (2014/694) and conducted in accordance with the Helsinki declaration. All participants signed an informed consent prior to enrollment and received a compensation of 500 NOK for their participation.

CCT protocol

The computerized working memory training program (Cogmed Systems AB, Stockholm, Sweden) consisted of 25 online training sessions. In this study, to increase feasibility, we utilized 17 sessions over a period of three to four weeks, corresponding to approximately five weekly training sessions (Kolskår et al., 2020; Richard et al., 2020). On average, patients completed two training sessions combined with tDCS (stimulation was applied during the first 20 min of the training session) per week with a minimum of one day between each tDCS session. The remaining CCT sessions were performed at home. Each training session comprised eight different exercises and lasted for about 45 min. In total, 10 different tasks targeting verbal and visuospatial working memory were used. The difficulty level of each task was automatically adapted to the participant’s performance throughout the intervention.

tDCS protocol

The details of the protocol have been described previously (Kolskår et al., 2020). Briefly, participants were randomly assigned to an active or a sham condition. We used a battery-driven direct current stimulator (Neuroconn DC-STIMULATOR PLUS, neuroConn GmbH, Illmenau, Germany), with the following parameters: DC current = 1 mA, total duration = 20 min, ramp-up = 120 s, fade-out = 30 s, and current density = 28.57 µA/cm2, rubber pads size = 5 × 7 cm. We used the factory settings for the sham condition, including a ramp-up and a fade-out period. Based on the 10–20 system for the electrode location, the anodal electrode was placed over F3 and the cathodal electrode over O2. The pads were covered with high-conductive gel (Abralyt HiCl, Falk Minow Services Herrsching, Germany) to keep the impedance threshold under <20 kΩ and fixated with rubber bands. Side-effects were monitored following each session through self-report forms.

TVA-based assessment

TVA-based modeling was based on data from a whole-report paradigm (Dyrholm et al., 2011; Sperling, 1960), in which six red letters from a set of 20 different letters (ABDEFGHJKLMNOPRSTVXZ) were briefly presented on a circle for either 20, 40, 60, 110, and 200 ms terminated by a pattern mask or presented for 40 or 200 ms unmasked. The paradigm was presented on a 24” BenQ XL2430T gaming monitor at a refresh rate set of 100 Hz. The participants were seated at a distance of 60 cm in a semi-dark room. Participants were instructed to report all the letters they were “fairly certain” of having seen (i.e., to use all available information but refrain from pure guessing). The paradigm comprised 20 practice trials and 140 test trials (i.e., 20 trials for each of the seven exposure duration conditions), with an overall duration of approximately 20–25 min, including the instruction and the practice trials. TVA parameters K, C, t0 were estimated by a maximum-likelihood procedure using the LibTVA toolbox (Dyrholm et al., 2011). The model had 8 degrees of freedom (df): K, 5 df (the value reported is the expected K given a particular distribution of the probability that on a given trial, K = 1, 2, …, 6); C, 1 df; t0, 1 df; and µ(additional effective exposure duration for unmasked letters), 1 df. For those participants whose t0 was estimated to be below 0, we refitted the data fixing t0 at 0. In addition, the error rate (i.e., the percentage of incorrect letters out of the reported letters) was calculated.

Processing of cogmed data

We used the same performance improvement scores as in previous publications (Kolskår et al., 2020; Richard et al., 2020). Briefly, we used linear modeling with performance as dependent variable and session as independent variable to quantify the changes in performance across the training period for each participant and for each trained task. Next, we performed a principal component analysis (PCA) on the performance improvement scores (zero-centered and standardized beta estimates from the linear models) and used the first factor as the individual’s performance improvement across the trained tasks.

Statistical analysis

Statistical analyses were performed using R version 3.3.3 (2017-03-06) (R Core Team, 2017). To test our hypothesis of impaired attentional functions in stroke patients compared to healthy peers, we compared the TVA parameters between groups using linear models with each of the TVA parameters as dependent variables, group (patients and controls) as independent variable, and age and sex as covariates. Since we observed a difference between groups in term of level of education, we performed an additional analysis including education in the model. To control for the number of tests, we employed Bonferroni correction with α = 0.05/4. Cohen’s D was calculated using two times the t-value divided by the square root of the degrees of freedom.

To test for associations between clinical characteristics and severity at hospital discharge and TVA parameters, we used linear models with each of the TVA parameters as dependent variables and each of the clinical scores (NIHSS, TOAST classification, lesion location) as independent variables, including age and sex as covariates in all models.

To assess the reliability of the TVA parameters in a longitudinal context, we estimated the intra-class coefficient (ICC) using ICCest function from the ICC R package (Wolak, Fairbairn & Paulse, 2012) across the six sessions.

To test if higher attentional performance at baseline (first main assessment) was associated with larger cognitive improvement during the course of the intervention, we conducted four linear models with Cogmed performance gain as dependent variable and each of the TVA parameters as independent variable, including age and sex as covariates in all models. As a supplemental analysis, we added tDCS group (sham vs experimental) as an additional variable and tested for interactions between tDCS and each of the TVA parameters on cognitive improvement.

To test whether improvement in TVA performance scores was associated with improvement in CCT performance, we did the following: first, we estimated TVA performance change using linear modeling with performance as dependent variable and session as independent variable to quantify the changes in performance across the six TVA assessments for each participant and for each TVA parameters. Then, we calculated the correlations between the beta estimates for the TVA parameters and the CCT performance score.

Lastly, we investigated whether tDCS condition was associated with improvement rate on TVA using linear mixed effect models (LME) testing for associations between TVA parameters and session (time) by group (active and sham).

Results

Cross-sectional – case-control

Table 1 shows the statistics for the case-control comparisons and Fig. 2 depicts the associations between age and each of the TVA parameters for the stroke patients and the healthy controls. Table 2 shows the summary statistics for the four linear models testing for association between TVA parameters and group (case-control comparisons). Briefly, at the group level, patients performed significantly poorer than healthy controls on short-term memory capacity (K), processing speed (C) and perceptual threshold (t0). Error rate did not significantly differ between groups. In addition, the analysis revealed significant main effects of age on parameters K, C and t0 indicating poorer performance for older participants and a significant main effect of sex on t0, indicating higher perceptual threshold in women, but no significant main effect of sex on K and C. Table S1 shows the summary statistics for the four linear models testing for association between TVA parameters and group (case-control comparisons) when adding education as a covariates and shows similar association patterns as described above.

Figure 2 Pearson correlations between age and TVA parameters for each of the two groups independently.

(A) Short-term memory capacity (K). (B) Perceptual processing speed (C). (C) Perceptual threshold (t0). (D) Error rate. HC: Healthy controls. IVS: Stroke patients.

Table 2 Summary statistics for the linear models testing for association between TVA parameters (dependent variables) and group (case-control comparisons).

Parameter	Group	Age	Sex	
K	estimate (std.error)	−0.334 (0.104)	−0.020 (0.005)	−0.138 (0.102)	
	t (p)	−3.207 (0.002)*	−3.910 (<.001)*	−1.349 (0.179)	
	Cohen’s d	−0.447	−0.545	−0.188	
C	estimate (std.error)	−6.585 (1.983)	−0.337 (0.099)	2.368 (1.951)	
	t (p)	−3.321 (0.001)*	−3.389 (0.001)*	1.214 (0.226)	
	Cohen’s d	−0.463	−0.472	0.169	
t0	estimate (std.error)	7.828 (2.435)	0.407 (0.122)	−6.684 (2.395)	
	t (p)	3.215 (0.002)*	3.334 (0.001)*	−2.790 (0.006)*	
	Cohen’s d	0.448	0.465	−0.389	
error rate	estimate (std.error)	−0.021 (0.013)	0.000 (0.001)	−0.003 (0.012)	
	t (p)	−1.676 (0.095)	−0.015 (0.988)	−0.273 (0.785)	
	Cohen’s d	−0.234	−0.002	−0.038	
Notes.

* Significant after Bonferroni correction.

K = short-term memory capacity; C = perceptual processing speed; t0 = perceptual threshold. Estimate = unstandardized regression coefficients b; std.error = standard error; t = t-value; p = p-value.

Cohen’s D was calculated using two times the t-value divided by the square root of the degrees of freedom.

Table 3 shows the summary statistics for the associations between each of theclinical measures and each of the TVA parameters. Briefly, we found no significant associations between clinical measures (NIHSS at hospital discharge, TOAST classification or stroke location) and parameters K, C, t0 and error rate.

Table 3 Summary statistics for the associations between each of the TVA parameters (dependent variables) and each of the clinical measures.

Parameter	NIHSS			TOAST	location	
	estimate (std.error)	t (p)	Cohen’s d	F (p)	F (p)	
K	−0.066 (0.063)	−1.037 (0.304)	−0.27	1.593 (0.188)	2.363 (0.08)	
C	0.408 (0.977)	0.418 (0.678)	0.109	1.937 (0.116)	0.756 (0.523)	
t0	1.997 (1.723)	1.159 (0.251)	0.302	1.188 (0.325)	0.740 (0.532)	
error rate	−0.003 (0.007)	−0.376 (0.708)	−0.098	0.310 (0.87)	1.603 (0.198)	
Notes.

* Significant after Bonferroni correction.

a Main effect of TVA parameters on Cogmed performance.

K = short-term memory capacity; C = perceptual processing speed; t0 = perceptual threshold; NIHSS = National Institute of Health Stroke Scale; TOAST = Trial of Org 10172 in Acute Stroke Treatment (TOAST) classification system; location = coarse four-class classification of lesion location (left hemisphere, right hemisphere, bilateral and brain stem/cerebellum lesions); Estimate = unstandardized regression coefficients b; std. error = standard error; t = t -value; p = p-value.

Cohens D was calculated using two times the tvalue divided by the square root of the degrees of freedom.

Longitudinal – reliability and change over time

Figure 3 shows individual performance for each of the TVA parameters across the six timepoints for each group (sham vs tDCS) together with the global inter-class coefficients (ICCs) ranging from .58 (95% CI [.47–.69]) for perceptual threshold to .80 (95% CI [.72–.86]) for short-term memory capacity.

Figure 3 Individual performance for each of the TVA parameters across the six timepoints for each group (sham vs tDCS).

Reliability of each TVA parameter is indicated by the intra-class coefficient (ICC) with 95% confidence interval (CI). TVA: Theory of visual attention. tDCS: transcranial direct current stimulation. (A) Short-term memory capacity (K). (B) Perceptual processing speed (C). (C) Perceptual threshold (t0). (D) Error rate.

Table 4 shows summary statistics from the four linear models testing for associations between Cogmed performance gain and TVA performance at baseline. Briefly, the analysis revealed one significant association between Cogmed performance gain and processing speed (C) suggesting that higher processing speed at baseline was associated with larger cognitive gain during the course of the intervention. Beyond this, we found no significant effect of age or sex on performance gain. The analysis including tDCS group (sham and experimental) as an additional variable revealed no significant main effect of tDCS group, nor tDCS group by TVA performance interaction on Cogmed performance gain.

Table 4 Summary statistics from the linear models testing for associations between Cogmed performance gain (dependent variable) and TVA performance at baseline, including age and sex as covariates.

Parameter	TVAa		Age		Sex		
	estimate (std.error)	t (p)	estimate (std.error)	t (p)	estimate (std.error)	t (p)	
K	−0.33 (0.304)	−1.07 (0.289)	0.01 (0.031)	0.30 (0.763)	−0.42 (0.507)	−0.84 (0.407)	
C	−0.08 (0.027)	−2.79 (0.007)*	−0.01 (0.030)	−0.46 (0.645)	−0.45 (0.476)	−0.95 (0.344)	
t0	0.02 (0.012)	1.35 (0.183)	0.01 (0.030)	0.37 (0.713)	−0.33 (0.512)	−0.64 (0.522)	
error rate	−3.21 (2.717)	−1.18 (0.242)	0.02 (0.031)	0.72 (0.476)	−0.36 (0.512)	−0.70 (0.488)	
Notes.

* Significant after Bonferroni correction.

a Main effect of TVA parameters on Cogmed performance.

K short-term memory capacity

C perceptual processing speed

t0 perceptual threshold

Estimate unstandardized regression coefficients b

std.error standard error

t t-value

p p-value

Table 5 shows summary statistics from the LME models testing for associations between TVA parameters and group (sham and experimental) by session interaction, including age and sex as covariates and participant as random factor. The models revealed robust main effect of session on each of the TVA parameters, suggesting improvement in performance over time. Error rate increased over time. Beyond this, the analyses revealed a main effect of age on processing speed (C). We found no group by session interactions on any of the TVA parameters. Figure 4 shows the correlations between performance improvement over the course of the CCT (Cogmed factor scores) and the change in TVA parameters across the six sessions (beta estimates). There was a positive correlation between improvement in CCT and K, as well as the error rate, and a negative correlation with t0, suggesting that participants showing a larger improvement in Cogmed also showed an increase in short-term memory capacity (K), made more mistakes across TVA sessions, and decreased their perceptual threshold (t0). The estimated change in speed of processing (C) across sessions did not show any significant correlations with Cogmed improvement.

Table 5 Summary statistics from the linear mixed effects models testing for associations between TVA parameters (dependent variables) and group by session, including age and sex as covariates and participant as random factor.

Term		K	C	t0	Error rate	
session	estimate (std.error)	0.090 (0.014)	2.004 (0.316)	−2.502 (0.427)	0.006 (0.002)	
	t (p)	6.620 (<.001)*	6.340 (<.001)*	−5.860 (<.001)*	3.180 (0.002)*	
age	estimate (std.error)	−0.029 (0.012)	−0.666 (0.169)	0.167 (0.248)	0.001 (0.001)	
	t (p)	−2.410 (0.019)	−3.940 (<.001)*	0.670 (0.504)	0.960 (0.344)	
sex	estimate (std.error)	−0.213 (0.208)	1.967 (2.921)	−1.147 (4.286)	0.007 (0.021)	
	t (p)	−1.020 (0.311)	0.670 (0.504)	−0.270 (0.79)	0.330 (0.744)	
group	estimate (std.error)	0.003 (0.194)	1.113 (2.994)	2.005 (4.301)	−0.024 (0.020)	
	t (p)	0.020 (0.987)	0.370 (0.712)	0.470 (0.643)	−1.170 (0.247)	
sess:group	estimate (std.error)	−0.018 (0.019)	0.787 (0.446)	−0.361 (0.603)	0.005 (0.003)	
	t (p)	−0.930 (0.352)	1.760 (0.079)	−0.600 (0.550)	1.730 (0.085)	
Notes.

* Significant after Bonferroni correction.

K short-term memory capacity

C perceptual processing speed

t0 perceptual threshold

sess:group session by group interaction

Estimate unstandardized regression coefficients b

std. error standard error

t t-value

p p-value

Figure 4 Associations between each of the TVA parameter improvement scores (beta estimates) and the Cogmed factor scores for each individuals (on the right).

(A) Correlations between performance improvement over the course of the CCT (Cogmed factor scores) and the change in TVA parameters across the six sessions (beta estimates). (B) The −log10 (p-value) matrix of the correlations between performance improvement over the course of the CCT (Cogmed factor scores) and the change in TVA parameters across the six sessions (beta estimates). (C) Associations between change in short-term memory capacity (K) (beta estimates) and the Cogmed factor score. (D) Associations between change in processing speed (C) (beta estimates) and the Cogmed factor scores. (E) Associations between change in perceptual threshold (t0) (beta estimates) and the Cogmed factor scores. (F) Associations between change in error rate (beta estimates) and the Cogmed factor scores.

Discussion

Attentional deficits following stroke are prevalent and pervasive, and are important predictors of functional and cognitive recovery. Here, we leveraged the computational framework provided by the TVA to demonstrate poorer storage capacity, lower processing speed and higher visual threshold in chronic stroke patients compared to age-matched healthy controls. Further, we demonstrated high reliability of the TVA parameters in stroke patients across six test sessions. We also showed that higher processing speed at baseline, as indexed by the C parameter from TVA, was associated with larger cognitive improvement during the course of cognitive training.

Based on the notion that TVA provides sensitive and specific measures of visual attention and that attentional deficits are common and pervasive following even relatively mild strokes, we first tested whether chronic stroke patients would show reduced performance compared to age-matched healthy controls in K, C and t0. Our results demonstrated that, at group level, patients who suffered mild to moderate stroke, defined here with a NIHSS score below or equal to seven at the hospital discharge, showed reduced performance compared to age-matched controls, suggesting that attentional deficits are present also in patients suffering from relatively mild stroke. These effects could not be explained by a difference in education level between groups, and are in line with previous literature showing that TVA is sensitive to subtle deficits (Bublak et al., 2005; Habekost & Bundesen, 2003; Habekost & Rostrup, 2006; Habekost & Rostrup, 2007). Next, to the extent that clinical characteristics and severity of the stroke at hospital discharge are sensitive to visual attention, we tested whether higher clinical burden was associated with lower performance on TVA as measured by K, C and t0. We found no associations between NIHSS at hospital discharge, suggesting that, among patients with relatively mild strokes, the severity of the stroke as measured by NIHSS is not a strong predictor of short-term memory capacity, processing speed, nor perceptual threshold as measured by TVA in a chronic stage. These findings are in line with a recent study reporting poor predictive value of NIHSS obtained at the hospital admission for cognitive functions in the acute phase (Abzhandadze, Reinholdsson & Sunnerhagen, 2020). Further, we found no significant association between TOAST nor location of the stroke, and performance on TVA. Thus, neither stroke severity as measured by NIHSS at hospital discharge, nor the etiology of stroke based on TOAST or stroke location provided predictive value of attentional performance measured by TVA in a chronic stage, highlighting the complexity of the clinical etiology of long-term attentional impairments following stroke.

In line with previous findings using three test sessions in healthy individuals (Habekost, Petersen & Vangkilde, 2014) and studies assessing internal reliability (Finke et al., 2005; Habekost & Rostrup, 2006), our results demonstrate highest reliability for the K parameter, and fairly good reliability for C and t0 across the six TVA sessions in chronic stroke patients. These findings are encouraging as they support both the feasibility and reliability of computational behavioral approaches in a clinical setting, and provide support for previous cross-sectional studies implementing a similar paradigm and modeling approach.

To test the extent that attentional abilities facilitate response to cognitive training, we tested whether higher attentional abilities as measured by the TVA at baseline were associated with larger improvement in response to cognitive training during the course of the intervention. Our results revealed a significant association between processing speed at baseline and performance improvement over the course of the intervention, indicating that patients with higher processing speed showed larger cognitive improvements, which is in line with the cognitive reserve hypothesis (Shin et al., 2020). In contrast, visual memory capacity and perceptual threshold at baseline were not significantly associated with cognitive improvement. A previous study in 68 healthy young individuals using three test sessions with one week interval reported significant improvement for processing speed, visual memory capacity and perceptual threshold, with the strongest practice effects for processing speed (Habekost, Petersen & Vangkilde, 2014). These previous findings are largely in line with our observations of substantial practice effect for all three TVA parameters across six test sessions in chronic stroke patients during the course of CCT. Future studies including appropriate control conditions are needed to assess to which degree cognitive training or other interventions influence TVA parameters beyond simple practice effects.

Further, we tested whether improvement in TVA performance was associated with improvement in CCT performance by correlating performance improvement over the course of the CCT (Cogmed factor scores) and the change in TVA parameters across the six sessions (beta estimates). The results suggested associations between both short-term memory capacity (K) and processing threshold (t0) and improvement in CCT performance, suggesting that whereas processing speed (C) was not significantly associated with CCT improvement, participants showing a larger improvement in Cogmed also showed an increase in short-term memory capacity (K), a decreased their perceptual threshold (t0), and an increased error rate across TVA sessions.

Our results jointly demonstrate that processing speed as indexed by the C parameter is both reliable, sensitive to case-control differences between stroke patients and age-matched healthy peers, and associated with response to cognitive training in chronic stroke patients. These findings support a clinical implementation of computational cognitive approaches in general and of TVA specifically in future stroke studies.

In line with one of the aims of the intervention, we also tested for effects of tDCS on cognitive improvement and interactions between attentional abilities as measured by TVA and tDCS on cognitive improvement. In addition, we tested for tDCS by time interaction on TVA performance to assess whether the experimental conditions would result in differential improvement in TVA performance over time. Here, corroborating previous publications using the same sample (Kolskår et al., 2020; Richard et al., 2020), we found no significant associations between tDCS group (sham and experimental) and cognitive improvement, nor tDCS group by session interactions on TVA performance, providing no support for a beneficial effect of tDCS. These results are in lines with a recent review and meta-analysis study reporting limited effect of tDCS in the context of memory training (Galli et al., 2019).

Several methodological considerations need to be highlighted while interpreting our results. First, as previously emphasized (Richard et al., 2020), patients included in this study represent a high functioning group with relatively mild cognitive deficits and presumably better prognosis compared to patients with more severe symptoms, limiting the generalizability of our findings. It is possible that the current analysis would have revealed stronger associations between clinical variables and TVA parameters in a sample including a broader range of stroke severities and cognitive impairments, or if using different stroke classification schemes. The labor-intensity of the current intervention and imaging study was demanding for the individual patient, preventing the study from sampling a wider range of the stroke severity spectrum. Future studies are needed to test if the value of TVA and other computational behavioral approaches may be particularly high in relatively well functioning samples where the cognitive deficits are assumed to be subtle (Ulrichsen et al., 2020). Moreover, the cross-sectional design prevents us from determining whether the group differences between patients and controls in TVA parameters primarily result from the stroke, or were present prior to the stroke. Further, the lack of control group performing an alternative or sham CCT in the longitudinal study does not allow us to separate beneficial effects of Cogmed on TVA performance from learning effects in response to repeated TVA assessments. Indeed, previous studies have reported significant practice effect on all three TVA parameters, with the strongest effect observed for processing speed (Habekost, Petersen & Vangkilde, 2014). Further, regarding the reliability estimates, although the reliability coefficients for all TVA parameters were fairly high and in line with previous studies, it is possible that the individual variability in the CCT gain could have led to underestimating ICC for the different TVA parameters. Thus, the reported values should be considered lower-bound estimates. Lastly, while none of the patients reported severe visual or linguistic impairments, future studies should apply relevant clinical assessments to rule out an impact of subtle visual or language deficits on task performance, which will be important to assess the validity of TVA performance as an indicator of attentional impairments.

Conclusions

In conclusion, we have assessed the sensitivity and reliability of TVA parameters assessing short-term memory capacity (K), processing speed (C) and perceptual threshold (t0) derived using a whole-report behavioral paradigm in a cross-sectional case-control comparison and longitudinal assessment during the course of a CCT scheme in chronic patients who suffered mild stroke. Our results demonstrate poorer K, lower C and higher t0 in chronic stroke patients compared to healthy controls. Further, we demonstrated high reliability of the TVA parameters in stroke patients across six test sessions, and showed that higher C at baseline was associated with larger cognitive improvement over the course of cognitive training. Thus, a clinically feasible implementation of TVA-based assessment offers sensitive and reliable computational parameters of short-term memory capacity, speed of processing and visual threshold in chronic stroke patients.

Supplemental Information

Supplemental Information 1 Summary statistics for the linear models testing for association between TVA parameters (dependent variables) and group (case-control comparisons), including age, sex and education as covariates

*Significant after Bonferroni correction. K = short-term memory capacity. C = perceptual processing speed. t0 = perceptual threshold. Estimate = unstandardized regression coefficients b. std.error = standard error. t = t-value. p = p-value. Cohen’s D was calculated using two times the t-value divided by the square root of the degrees of freedom.

Click here for additional data file.

Additional Information and Declarations

Competing Interests

Author Contributions

Human Ethics

Data Availability

Jan Egil Nordvik is employed by CatoSenteret Rehabilitation Center. All other authors declare that they have no competing interests.

Geneviève Richard conceived and designed the experiments, performed the experiments, analyzed the data, prepared figures and/or tables, authored or reviewed drafts of the paper, and approved the final draft.

Anders Petersen analyzed the data, authored or reviewed drafts of the paper, and approved the final draft.

Kristine Moe Ulrichsen performed the experiments, authored or reviewed drafts of the paper, and approved the final draft.

Knut K. Kolskår conceived and designed the experiments, performed the experiments, authored or reviewed drafts of the paper, and approved the final draft.

Dag Alnæs conceived and designed the experiments, authored or reviewed drafts of the paper, and approved the final draft.

Anne-Marthe Sanders performed the experiments, authored or reviewed drafts of the paper, and approved the final draft.

Erlend S. Dørum conceived and designed the experiments, performed the experiments, authored or reviewed drafts of the paper, and approved the final draft.

Hege Ihle-Hansen analyzed the data, authored or reviewed drafts of the paper, interpretation of the data, and approved the final draft.

Jan E. Nordvik conceived and designed the experiments, authored or reviewed drafts of the paper, research infrastructure, and approved the final draft.

Lars T. Westlye conceived and designed the experiments, analyzed the data, authored or reviewed drafts of the paper, research infrastructure, and approved the final draft.

The following information was supplied relating to ethical approvals (i.e., approving body and any reference numbers):

The study was approved by the Regional Committee for Medical and Health Research Ethics (South-East Norway, 2014/694).

The following information was supplied regarding data availability:

The data is available at OSF: Richard, Genevieve, and Lars T Westlye. 2019. “TVA-Based Modeling of Short-Term Memory Capacity, Speed of Processing and Perceptual Threshold in Chronic Stroke Patients.” OSF. August 26. osf.io/wu97n.

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
