# Peer review of "TVA-based modeling of short-term memory capacity, speed of processing and perceptual threshold in chronic stroke patients undergoing cognitive training: case-control differences, reliability, and associations with cognitive performance"

_PeerJ, doi:10.7717/peerj.9948_

## Round 0.1 · original submission · Major Revisions

Your manuscript has been evaluated by three independent reviewers. Though they have a favorable view of your paper, they also recommended a substantial review of the original manuscript. Some concerns about the lack of a control arm in the longitudinal intervention were also pointed out.

·

Basic reporting

Review of manuscript number (#40638)
TVA-based modeling of short-term memory capacity, speed of processing and perceptual threshold in chronic stroke patients: Case-control differences, reliability, and sensitivity to cognitive training
By Richard et al.

Summary

The authors describe a combination of a case-control experiment and a training intervention study in chronic, mild stroke patients. They used a test based Bundesen’s theory of visual attention (TVA) to derive estimates of visual attention at the individual level to investigate if the patient group differed from a matched control group at baseline and if improvements in cognition following a computerised cognitive training intervention across several weeks in the patient group could be linked to their baseline attentional performance. Orthogonal to the cognitive training, patients participated in a tDCS protocol with half of the participants receiving active tDCS stimulation and half receiving sham stimulation. Additionally, the reliability of the attentional parameters derived from TVA-based testing was assessed in the patient group through repeated measures.
The authors found lower short-term memory capacity (TVA parameter K), slower perceptual processing speed (TVA parameter C), and an elevated perceptual threshold (TVA parameter t0) in the case-control comparison, indicating attentional sequelae even after mild stroke.
The longitudinal analyses linked to the training intervention revealed fair to good reliability of TVA-based attentional estimates and that higher processing speed before commencing training was associated with higher intervention gains.
The work undertaken to conduct this study is very impressive and the inclusion of chronic patients strengthens the results markedly.


Major comments

Introduction
While the interdiction is clearly written, it can be improved in both the description of the groups and of the hypotheses and their relative importance. In Hypothesis 1 (from line 91) there is no distinction between the three different TVA-based parameters. Does this mean that they are equally likely to be affected by stroke? Similarly, for Hypothesis 2, do you expect all parameters to be affected? As part of the test for associations between stroke subtype and TVA parameters, do you have any particular hypotheses? Also for Hypothesis 4, there is no distinction between the parameters. Does this mean that you hypothesise that they are equally predictive for larger cognitive improvement. Finally, in Hypothesis 3, it is hypothesised that TVA parameters will provide reliable and sensitive measures of specific attentional functions. While you test for reliability, it is unclear to me how you operationalise and test sensitivity.
At the end of the introduction, the use of tDCS as an adjunct intervention is presented, however, it is not clear whether any specific hypotheses are linked to this part of the intervention. Here it is also a bit unclear if the two patient groups are actually overlapping of distinct.

Materials and methods
Again, it is unclear from the text if the patients included in the two parts of the study are the same. If they are partly overlapping, what characterised the patients who dropped out?
In line 85, the patients are described as having a NIHSS score below 7, however Table 1, the range of NIHSS scores go from 0-7. Were patients with a score = 7 included or not? And how was <7 chosen as a cut-off for mild stroke, when this is in the lower range of “moderate stroke”. You might want to consider using another reference than Lyden et al. (2009) for this.

Table 1; Please state and explain all abbreviation in the table text.

If Figure 1 is to be used to support the understanding of the study flow, it should be improved and integrated more with the text; two different baseline assessments are shown, however, the rationale for this is not described in the text, nor does it comment on the waiting period. More importantly, even though baseline performance is hypothesised to be associated with cognitive improvement from training, it is never stated in the entire manuscript which of the two baselines this refers to (e.g., line 208). Finally, if Figure 1 is to aid my understanding of the flow, it should also depict the tDCS intervention. As a minor thing, please also write out abbreviations in the figure text.

In the description of the patient sample, I was surprised that there was no assessment of nor mentioning of visual impairments. I find it unlikely that none of the patients had visual impairments that could interfere with their performance in a visual attention test. This should be addressed and discussed.

Line 139; “three cognitive assessments” - are these different from the TVA-based assessments and how?

TVA-based assessment; Thank you for clearly describing both the test and fitting procedure/model. In Table 1, the range for t0 is stated to be from 0 and up. This indicates that fits with negative t0-values, were refitted using 0 as a lower bound. Please state if this is the case.

Processing of Cogmed data and statistical analysis; The processing is elegantly done, but I would like the explanation to be a bit more elaborate. Could you state clearly that it is the beta-estimates (e.g., slopes) for each test and for each participant that are employed? Also, it may aid the understanding if you refer to the “common scores” as “factor scores”.
It would be instructive to know, how you calculated the Cohen’s ds that are used and what you consider to be the range for “good ICC”. There are different views on this, which directly impact how your results should be interpreted.

Results
In general, the results are reported in an orderly and intuitive way. I would suggest not write out the adjusted p-value, as what you adjust is the alpha-value and the additional text adds to the slight cluttering of the results. There is a marked shift between the number of results reported in the Cross-sectional/case-control section and in the Longitudinal section. I would suggest balancing this better.

Discussion
Overall, I find the discussion to be a bit short, of which a lot of time is spent overviewing/iterating the different results without discussing them against previous literature. As an example lines 283-287 are stating almost exactly the same as the first section of the discussion, and lines 296-299 are also an iteration. In relation to this, I do not see how the two null-findings highlight “…the complexity of the clinical etiology of long-term attentional impairments following stroke”. As a general comment, there is a general lack of references to studies conducted by other research groups. Likewise, more discussion on cognitive training in rehabilitation of stroke, would bolster the discussion further, as would considerations of the impact of age (which is also alluded to in your introduction).

It is nice to see (at least two) links to previous TVA-based studies of stroke patients and reliability of the estimates. I would, however, like to see a discussion of the size of the effects/the reliability. Are the same attentional parameters affected in other studies (there are a wide range of stroke studies with TVA-based measures available). Also, I would like to know if there are links between the Cogmed performance increase and the increase in TVA-based parameters across sessions, i.e., the possible transfer effects from the computerised training to the functions that may underlie performance (according to your own hypotheses).

Line 319 reads, “In line with one of the major aims of the study, we also tested for effects of tDCS on cognitive improvement…”. Going back to the introduction, where the tDCS-intervention was only introduced as some kind of add-on with no specific hypotheses related to it, I think that it is unclear what the role of the tDCS manipulation is. Is it really a major aim?
In the following it is stated that using the same sample, no differential effects of tDCS were found in previously. It is not clear if you have already investigated some of the same things that are reported here or if it is only the effect of tDCS on other cognitive tests that were previously investigated? More importantly, it is imperative that you discuss these (null-)findings with reference to other studies that the ones, you have conducted yourselves on the same group.

I appreciate the discussion around both the generalisability of the use of computational testing to more severely affected stroke patients and the possible limitations pertaining to training effects in the TVA parameters. Specifically, it is a worthwhile consideration that this is not a “pure reliability” study, as the ICC of the TVA parameters are could be affected by individual variations in the response to the training intervention. As stated above, an investigation of the simultaneous increase/change in Cogmed scores and attentional abilities, may help elucidate this question.

Experimental design

No comment

Validity of the findings

No comment

Additional comments

Minor comments
I would suggest going through the manuscript with an eye for sentence length. Some examples include lines 36-41; 82-88.

Consider removing “including social interactions.” (line 30), as this is not assessed nor discussed at any later point.

Consider rephrasing the sentence covering lines 303-306, as this appears unnecessarily convoluted.

Table 2; Please write out abbreviations + parameter names and make sure to change “Table 1” in the table text to Table 2.

Table 3; Please write out abbreviations + parameter names.

Reviewer 2 ·

Basic reporting

The paper is very well written. It is generally clear, follows a logical order and appears to be technically sound. The one area of ambiguity surrounds the use of the term “higher processing speed” (see for example line 258) I think it would be clearer to the reader if the term “faster processing speed” were used.
The introduction covers many of the issues the authors are trying to address with their study. The authors should make it clearer in the introduction that a number of publications have previously examined the use of TVA in stroke populations (for example, Habekost and Rostrup 2006, Peers et al 2005, Bublak et al 2005) and explain how this investigation adds this literature.
The paper would benefit a re-working of the hypotheses section, for example, it is unclear from the introduction what predictions are being made about stroke sub-type (as assessed by the TOAST) other than it might be interesting to see. Perhaps you could explain based on previous literature what you might have expected to find. It would be good for further clarification as to what is meant by hypothesis 3 (that TVA parameters would constitute reliable and sensitive measures of attentional functions), being that the longitudinal study includes a cognitive intervention. It seems that if you are testing the hypothesis in this design then you might expect that if the parameters are sensitive that they should be able to pick up subtle changes in cognitive function over the course of the intervention. However reliability should be measured by consistency in the TVA parameter estimates at each time point. Without a waitlist control arm where patients complete the TVA assessments without any intervening intervention it is unclear to me how this hypothesis can be addressed.
It is clear that a lot of work has been undertaken to collect an important substantial amount of both cross-sectional and longitudinal data from patients, and in the case of the cross-sectional data from healthy controls as well. It is clear that data other aspects of this project has previously been published (Kolskar et al 2019, Richard et al 2019, Ulrichsen et al 2019). The authors need to make it clearer how the cross-sectional aspects and longitudinal aspects of the data fit together to a single cohesive piece rather than a series of interesting but only loosely connected hypotheses.
The article is well structured this Figures and tables are appropriate and relevant to the points the authors wish to make.

Experimental design

The experiment fits within the aims and scope of PeerJ. As mentioned in the previous section the authors need to be clearer in the introduction as to how this project aimed to replicate or extend the existing findings in the field (particularly in relation to previous TVA stroke studies, see above , which are currently not discussed in detail). Currently the hypotheses section provides a set of seemingly disparate hypotheses, it would be good for the authors to re-work this in to a single overarching question, and then state how their existing hypotheses fit in to this central question.
The study was carried out to a high ethical standard and appropriate measure were used to measure TVA parameters. It is unclear whether measures such as the TOAST and NIHSS would be sensitive enough for the purpose of measuring the effects of severity or type of stroke on TVA performance, it would be useful for the authors to comment on this. Furthermore the methods would benefit from an expanded explanation as to why the standard protocol was not used for the cogmed training(17 days rather than the standard 20-25 days) , and how and how and why there was deviation in the number of days between TDCs. I do, however, have serious concerns about the design of the longitudinal intervention study as it lacks a waitlist control arm which is critical in assessing the sensitivity and reliability of both the intervention (and more importantly in the case of this paper the TVA parameters). This waitlist arm would be key in determining whether TVA is sensitive to measuring any form of change or whether any change measured is simply down to practise effects of repeated exposure to the tasks. One way to potentially address this point would be to look at the amount of progress made on the Cogmed training tasks was associated with the extent of changes within the TVA parameters. On a related note please could you provide some details in your results section about improvements on the cogmed tasks over the course of the intervention.
The methods are described in a suitable amount of detail to allow for replication

Validity of the findings

The authors have conducted a very large study assess of stroke patients, it would be great if they could provide a more detailed description in the discussion as to how these fit with previous findings of TVA assessment in stroke (for example, Habekost and Rostrup 2006, Peers et al 2005, Bublak et al 2005). The authors are clear in pointing out the limitations in their findings, however as I have previously noted the lack of a control arm in the longitudinal intervention arm really limits the ability to draw meaningful conclusions about the sensitivity and reliability of the measures. I do feel this situation could be potentially improved by examining the link between improvements on the cognmed tasks and changes in specific TVA parameters

·

Basic reporting

The paper is clearly written and conforms to professional standards. The “Introduction” section provides a solid context. Nevertheless, I have some concerns:

- I think you could improve on explaining what exactly is assessed by the whole report task. In which way do the three TVA-parameters - threshold, speed, and VSTM capacity
– represent attentional abilities? In the “Introduction” section, several different abilities are mentioned, e.g. sustaining and focusing, which are quite different in neuro-cognitive terms. In particular, you talk about selection between competing stimuli (line 54), or between relevant and irrelevant information (line 69). As a result, it becomes not quite clear to which of these different aspects of attention the whole report task refers to. All the more, when considering the fact that the whole report task does not include any irrelevant stimuli. Perhaps, you might consider referring to a concept of attentional capacity which comprises speed and storage aspects, and is a core determinant of cognitive performance in general.

- lines 52/53: The background could be strengthened by providing a reference for “the distributed functional neuroanatomy supporting the range of attentional sub-functions”; see e.g. http://dx.doi.org/10.1016/j.tics.2017.01.011

- lines 78/79: Please mention that the letter displays are only presented for very brief exposure durations.

- lines 80/81: I am not sure whether the reader not familiar with TVA is able to understand the core idea of TVA from a single sentence. Thus, you should provide some more information on the TVA basics.

- line 83: Please mention already at this point, that the CCT was combined with tDCS. Otherwise, the later appearance of tDCS as a procedure in line 108 is extremely puzzling. Also include tDCS in Figure 1.

-Table 1: the abbreviation “lm” (case-control-comparison for the TVA parameters) is not explained. Presumably, it means “linear modeling”. Please specify!

Experimental design

- The rationale behind your fourth hypothesis (line 103: higher attentional abilities at baseline associated with larger cognitive improvement during CCT) requires some explanation. After all, lower levels of ability might, in principle, leave more room for improvement, which would predict that lower levels take more gain from training. Perhaps, your hypothesis is related to the concept of “cognitive reserve”?

- Please provide details of the experimental set-up and procedure! For example, did the experiment run PC-controlled, in a darkened room? What was the viewing distance? Size of the stimuli? Duration of the assessment? Etc.

- The statement from the discussion section (334/335: constraints for patient recruitment) should be included into the description of the patient sample.

- Were basic visual functions like visual acuity and contrast sensitivity assessed? If not, this should be mentioned as a limitation.

- Also, there is no information on psychiatric disorders in patients, in particular symptoms of anxiety and depression. This should also be considered as a limitation.

Validity of the findings

I thank the authors for their clear presentation of the methods and results. However, some points should be considered:

- While the healthy control group was matched by age and sex, this was not done with respect to education, and mean educational level was significantly higher in the healthy control group. In the statistical analysis, you used age and sex as covariates. The reason for this is not immediately clear to me, given the fact that the groups to be compared do not differ with respect to these variables. In contrast, although there was a significant difference with respect to education, this relevant variable was not included as a covariate in the analysis. Why is that? I would recommend repeating your analyses with education included as a covariate to see whether the results were affected by this group difference.

- I am not sure whether Figure 2 is really necessary: It is unclear how the association between age and TVA parameters relates to the hypotheses of the study. The case-control-comparison of TVA parameters is based on a group comparison, and both groups were well matched for age, as stated above. And for the longitudinal analysis for assessment of reliability, age also seems not to be a relevant factor. So, I would suggest skipping Figure 2 and the related statistics.

- There is no information on language disorders or visual field defects in the patient sample, whether these were criteria for exclusion or not. These are relevant issues in stroke patients. Please provide more information so that both the clinical feasibility of the TVA based assessment, and the validity of the findings as true attentional disturbances can be better assessed.

- With respect to the study aim of specific assessment of attentional sub-components, it appears as a problem that all three TVA-based parameters were affected. Does that mean that the different parameters are more or less indistinguishable indices of the same attentional deficit? Please provide the parameter inter-correlations, whether they are sufficiently independent from each other statistically.

- Patients were significantly impaired with respect to standard cognitive screening instruments (MMSE, MoCA). Therefore, the patients cannot be said to represent a “high-functioning group with relatively mild cognitive deficits” (line 330). Rather, it was quite a heterogeneous group with respect to cognitive impairment, with some patients severely impaired. You should also avoid concluding that TVA testing is “sensitive to subtle deficits” (line 289), because the data are not appropriate to support such a conclusion. It would, however, be interesting to see whether TVA parameters are related to global cognitive status, and, thus, reflect the degree of cognitive impairment to some degree. So, please provide correlations between parameters and cognitive screening instruments.

- In the “Discussion” section, you mention that baseline processing speed was related to “larger cognitive benefits of cognitive training” (lines 280, 312, 357). Such a statement is not correct, because there was no control condition for cognitive training, and the increase in processing speed might just be the result of several repetitions of the whole report. What can be stated is that parameter C appears to somehow indicate larger plasticity. Please revise!

- In line 299, you suggest that the failure of NIHSS and TOAST scores for explaining some of the parameter variance might be due to the complexity of the clinical etiology of attentional impairments after stroke. However, you should consider the possibility that associations would be revealed, if a different stroke classification was applied: For example, a differentiation according to the main cerebral arteries involved (anterior, middle, and posterior cerebral arteries).

- Lines 319-327: tDCS was not an a priori hypothesis, and also does not provide any additional information relevant for the scope of the paper: So; I would recommend skipping this paragraph.

- At the end of the “Discussion” section, please elaborate a bit on the implications for the clinical application of a TVA-based assessment in stroke patients! I think, this issue is under-represented. For example, what can the TVA-based assessment achieve that cannot be accomplished by cognitive screening procedures? And do these possible advantages justify the extra time required for assessment?

---

## Round 0.2 · Major Revisions

Your manuscript was re-reviewed by the original reviewers and one of the reviewers recommended acceptance and the other rejection. As you can see, the major concerns of reviewer 2 are related to experimental design and the conclusions made by the authors from this data. As stated by the reviewer: "I am concerned about the strong claim you make about sensitivity as I am not convinced that this can be addressed adequately with the current data." Thus, I realize that you can change the tone of your conclusions and highlight the points raised by the reviewer ("the lack of a control arm really limits any conclusions on the sensitivity of TVA to measure the change in patients.)" as a limitation of the study. If you are willing to do this, I will be glad to accept your manuscript. Please, also consider the inclusion of 10.1080/09602011.2018.1554534 in your manuscript.

·

Basic reporting

The authors have done a great job improving the manuscript and clearing up any ambiguities or lack of reporting that were pointed out in the review.
Also, revised and new versions of figures and tables add significantly to the readability of the article.

Experimental design

The experimental design has been described more clearly and the hypotheses now appear relevant and aligned with the design and the possibilities/limitations afforded by this. At the same time, it is now clear that the study addresses a knowledge gap in the literature.

The suggestions for improving the analyses have all been incorporated and the results and discussion sections have benefitted substantially from this.

Validity of the findings

I only have two minor comments:

1) I was happy to see that the authors included a remark referring to none of the patients as having severe visual or linguistic impairments in both the methods section and in the discussion. In the latter it was mentioned as a potential limitation that this was not formally assessed. To aid the reader in understanding the possible impact of e.g. visual impairments in the current study, I would like to see a single sentence on the validity of the findings as attentional impairments and not "just" visual defects.

2) In the discussion it is stated "Further, the lack of control group in the longitudinal study does not allow us to separate the potential benefits from CCT on attention from the learning effect from repeated TVA assessment". I would encourage the authors to discuss whether the putative benefits of CCT are over and above the improvements to be expected from repeated TVA-based assessment, based on previous studies looking at this.

Reviewer 2 ·

Basic reporting

The paper now reads well and the authors have worked hard to incorporate the previous literature into this version of the manuscript.
You may wish to include this paper as a previous study using TVA to measure training effects in stroke patients Peers et al. Neuropsychol Rehabil. 2020;30(6):1092-1114. doi:10.1080/09602011.2018.1554534
The paper is well structured and figures and tables are clear.
The hypotheses have been re-worked and are now clearer

Experimental design

I am still very concerned about the experimental design and the conclusions you draw from this data. Whilst great efforts have been taken to collect longitudinal data from a good sample of stroke patients whilst they complete cognitive training, the lack of a control arm really limits any conclusions on the sensitivity of TVA to measure change in patients. I am pleased to see that you attempted to examine this issue more closely by looking to see whether change in performance on CCT was associated with specific changes in TVA parameters. Your analysis suggesting that improvements in CCT were associated with increased K, decreased perceptual thresholds, but also increased errors, suggests to me that you are measuring a change in speed accuracy trade-off rather than a more profound change in cognitive function as a result of an intervention. I feel it would be better for you to focus the conclusions on reliability rather than discussing sensitivity as I really don't think it can be adequately addressed without the control arm.

Validity of the findings

I think this is well conducted work and if you were to simplify the message to indicate that TVA is reliable measure and is feasible for use in studies in patients with a range of severity then it is important work. I am concerned about the strong claim you make about sensitivity as I am not convinced that this can be addressed adequately with the current data.

---

## Round 0.3 · accepted · Accept

Thank you for revising your manuscript according to the suggestion of the reviewer. I realize that the majority of the points raised by the reviewer were considered and applied to the revised manuscript.